# Adipose Tissue-Derived Mesenchymal Stromal/Stem Cells, Obesity and the Tumor Microenvironment of Breast Cancer

**DOI:** 10.3390/cancers14163908

**Published:** 2022-08-12

**Authors:** Andreas Ritter, Nina-Naomi Kreis, Samira Catharina Hoock, Christine Solbach, Frank Louwen, Juping Yuan

**Affiliations:** Obstetrics and Prenatal Medicine, Gynecology and Obstetrics, University Hospital Frankfurt, J. W. Goethe-University, Theodor-Stern-Kai 7, D-60590 Frankfurt, Germany

**Keywords:** ASCs/MSCs, obesity, breast cancer, tumor microenvironment, cancer-associated fibroblasts, cancer-associated stem cells, epithelial–mesenchymal transition, therapy resistance

## Abstract

**Simple Summary:**

Adipose tissue is the major microenvironment of breast cancer. Adipose tissue-derived mesenchymal stromal/stem cells (ASCs/MSCs) are key players in adipose tissue. ASCs/MSCs, particularly in the obese state, are critical in remodeling the tumor microenvironment and promoting breast cancer progression. In this review, we have addressed the impact of obesity on ASCs/MSCs, summarized the crosstalk between ASCs/MSCs and breast cancer cells, discussed related molecular mechanisms, and highlighted related research perspectives.

**Abstract:**

Breast cancer is the most frequently diagnosed cancer and a common cause of cancer-related death in women. It is well recognized that obesity is associated with an enhanced risk of more aggressive breast cancer as well as reduced patient survival. Adipose tissue is the major microenvironment of breast cancer. Obesity changes the composition, structure, and function of adipose tissue, which is associated with inflammation and metabolic dysfunction. Interestingly, adipose tissue is rich in ASCs/MSCs, and obesity alters the properties and functions of these cells. As a key component of the mammary stroma, ASCs play essential roles in the breast cancer microenvironment. The crosstalk between ASCs and breast cancer cells is multilateral and can occur both directly through cell–cell contact and indirectly via the secretome released by ASC/MSC, which is considered to be the main effector of their supportive, angiogenic, and immunomodulatory functions. In this narrative review, we aim to address the impact of obesity on ASCs/MSCs, summarize the current knowledge regarding the potential pathological roles of ASCs/MSCs in the development of breast cancer, discuss related molecular mechanisms, underline the possible clinical significance, and highlight related research perspectives. In particular, we underscore the roles of ASCs/MSCs in breast cancer cell progression, including proliferation and survival, angiogenesis, migration and invasion, the epithelial–mesenchymal transition, cancer stem cell development, immune evasion, therapy resistance, and the potential impact of breast cancer cells on ASCS/MSCs by educating them to become cancer-associated fibroblasts. We conclude that ASCs/MSCs, especially obese ASCs/MSCs, may be key players in the breast cancer microenvironment. Targeting these cells may provide a new path of effective breast cancer treatment.

## 1. Introduction

The prevalence of obesity has tripled during the last decades, posing a major challenge to the entire society and health care systems worldwide [1,2,3]. Obesity, a condition of increased adiposity resulting from an imbalance between food intake and energy expenditure [4], is categorized according to body mass index (BMI) ≥30 kg/m^2^ [5]. Obesity associates with multiple disorders, including diabetes, hypertension, and cardiovascular diseases [6]. In addition, it is characterized by an increased incidence of cancer in various organs, such as the colon, rectum, kidney, pancreas, gallbladder, liver, thyroid, breast, ovary, and endometrium [7,8,9]. Obesity associates with inflammation and metabolic dysfunction, which greatly promote cancer development. Among various adipose tissue cell types, adipose tissue-derived mesenchymal stromal/stem cells (ASCs), belonging to mesenchymal stromal/stem cells (MSCs), are dramatically altered during obesity progression [10]. Obesity-associated ASCs are of crucial importance, contributing to the establishment of the breast cancer microenvironment and promoting the progression of breast cancer. In this narrative review, we have summarized the current knowledge regarding the potential pathological roles of obese ASCs/MSCs in the development of breast cancer, addressed the possible impact of obesity on ASCs/MSCs, discussed the intertwined relationship between ASCs and breast cancer cells, explored associated molecular mechanisms, and highlighted related research perspectives.

## 2. Method

A search was performed for original articles and reviews published between January 2000 and April 2022 in PubMed with a focus on ASCs/MSCs and breast cancer by using the following search terms (or combination of terms): ASCs, MSCs, obesity, breast cancer, cancer microenvironment, proliferation, invasion, metastasis, cancer progression, and therapy resistance. Only English-language and full-text articles were included.

## 3. Obesity and Breast Cancer

Among the malignancies, breast cancer is the most frequently diagnosed cancer and a common cause of cancer-related death in women. It originates from deregulation of normal growth pathways in mammary epithelial cells due to genetic mutations or epigenetic modifications [11]. Breast cancer is composed of distinct subtypes and is highly heterogeneous in both molecular and clinical terms. Based on the expression of the estrogen receptor (ER), progesterone receptor (PR), human epidermal growth factor receptor 2 (HER2), as well as a proliferation marker Ki67, breast cancer can be divided into four intrinsic molecular subtypes: luminal A (ER^+^, PR^+^, HER2^−^, Ki67^Low^), luminal B (ER^+^, PR^+^, HER^+/−^, Ki67^High^), HER2-enriched (ER^−^, PR^−^, HER2^+^), and basal-like subtype (ER^−^, PR^−^, HER2^−^), which largely resembles triple-negative breast cancer (TNBC) and comprises approximately 15% of all breast cancer cases [12]. Among these subtypes, TNBC, typically more aggressive, is associated with a higher rate of relapse and poor prognosis owing to the development of metastases in distant organs such as brain, liver, bone, and lungs [13,14]. The therapy of breast cancer consists of surgical resection, radiation, chemotherapy, and other therapeutic options, including antihormone therapy, signaling pathway targeting, DNA repair inhibition, aberrant epigenetic reversion, and immuno-oncology therapeutics [15]. Despite advanced therapeutic options, breast cancer remains one of the most common causes of female death [16], largely due to therapy resistance and metastases to brain, liver, lung, or bone. It is therefore necessary to explore the molecular mechanisms of cancer progression and pave new ways that offer more effective breast cancer treatment. 

Obese women with breast cancer have larger tumors and an enhanced risk of metastasis that contributes to a 30% increased risk of death [17,18,19]. Specifically, obesity associates with enhanced risk of more aggressive breast cancer as well as reduced survival of postmenopausal breast cancer patients [20,21]. While numerous meta-analyses have consistently shown positive associations between obesity and risk of hormone-receptor-positive (ER^+^ and PR^+^) breast cancer [22,23,24], growing evidence suggests that abdominal obesity, also known as central obesity, may increase the risk for TNBC in premenopausal women [25,26,27]. Moreover, compared to lean patients, obese patients respond worse to therapy, particularly when diagnosed with TNBC, contributing to the overall worse prognosis [27].

The dramatic increase in the prevalence of obesity, combined with the fact that over 75% of new cases of breast cancer occur in postmenopausal women [28], represents a pressing challenge in breast cancer prevention, treatment, and survival. Despite being a field of intensive research, the molecular mechanisms underlying the association of obesity with breast cancer are still incomplete.

### Adipose Tissue: Microenvironment, ASCs and Obesity

Although breast cancer initiation is largely driven by acquired genetic alterations, the tumor microenvironment (TME) is crucial in its progression. Breast cancer cells are mainly surrounded by mammary adipose tissue and intermingled with a repertoire of stromal cells such as ASCs, fibroblasts, endothelial, and immune cells. Breast cancer is able to change the adjacent adipose tissue by stimulating the transcription of genes associated with tumor growth, stemness, and progression [29,30]. In turn, adipose tissue cells, both adipocytes as well as stromal cells, promote cancer progression by secreting growth factors, cytokines, chemokines, and pro-migratory extracellular matrix (ECM) components [31,32]. Thus, breast-cancer-educated adipocytes and stromal cells, together with soluble factors as well as insoluble ECM proteins secreted by adipocytes, stromal cells, and cancer cells, constitute the TME [15,33]. Of importance, the TME is dynamic, as a multitude of stromal cell types, such as endothelial progenitors, immune cells, fibroblasts, ASCs, and MSCs, are recruited to develop the TME [34,35]. The TME is not only critical for cancer progression, but also for coordinating cancer plasticity and immune escape [15].

Obesity changes the landscape of adipose tissue. In the early stages of adipose tissue expansion, adipocyte hypertrophy generates a local hypoxia that contributes to increased secretion of adipokines, inflammatory cytokines, and lipid metabolites. These changes further reduce the metabolic flexibility of adipocytes, increase the rate of apoptosis, and recruit more inflammatory cells, including lymphocytes and macrophages, in adipose tissue [36]. The chronic hypoxia observed in obese adipose tissue results in chronic inflammation, endoplasmic reticulum (ER) stress, and an alteration of the ECM. In fact, obese adipose tissue is characterized by elevated saturated fatty acids released by obesity-associated lipolysis that induce macrophage activation via toll-like receptor 4 (TLR4) stimulating nuclear factor kappa B (NF-kB) signaling and inflammation in adipose tissues [37]. This further activates the transcription of pro-inflammatory genes including interleukin 6 (IL6), IL1β, and tumor necrosis factor α (TNFα) in adipocytes and stromal cells, causing local and systemic inflammation [38]. Accordingly, obesity is characterized by increased levels of circulating factors including insulin, insulin-like growth factor 1 (IGF-1), leptin, and inflammatory cytokines such as IL6 and TNFα [39]. Obese adipose tissue is further marked by increased infiltration of immune cells, cellular stress, hypoxia, insulin resistance, glucose intolerance, adipocyte hyperplasia and hypertrophy, reduced angiogenesis, and impaired tissue homeostasis [10]. In particular, obesity increases the number of myofibroblasts in mammary adipose tissue that deposits a more fibrillary and stiffer ECM, correlating with the malignant behavior of mammary epithelial cells and with inflammation [40]. Thus, various cell types of obese adipose tissue communicate with breast cancer cells, promote breast cancer progression, reshape metabolism in the TME, and suppress anti-tumor immunity [41,42,43]. Collectively, obesity changes the composition, structure, and function of adipose tissue, which associates with inflammation and metabolic dysfunction and promotes breast cancer development. 

Interestingly, adipose tissue, a mesodermal-derived organ, is the richest source of MSCs in the human body [44]. MSCs are multipotent cells involved in the maintenance of tissue homeostasis, regulation of local immune response, and regeneration of damaged tissues [45]. ASCs, like other MSCs in the body, have many characteristics in common with MSCs, including morphology, extensive proliferation potential, and the ability to undergo multi-lineage differentiation in vitro [32,46,47,48]. ASCs are able to produce a large variety of growth factors and have immunomodulatory properties [10,49,50]. Moreover, ASCs are anti-inflammatory because of their multiple capabilities, such as inhibiting the activation of natural killer cells, impairing cytotoxicity processes, reducing the proliferation of B cells, decreasing immunoglobulin production, and suppressing B cell functions [49]. 

However, obesity alters most of the enumerated functions of ASCs, which in turn affect their surrounding cells in adipose tissue [10,51]. In this context, the interaction between ASCs and breast cancer cells could have a crucial role in malignant progression [52,53,54]. In fact, ASCs have been shown to impact breast cancer progression by secreting cytokines/chemokines and other regulatory factors influencing angiogenesis, migration, and invasion of breast cancer cells [53,55,56]. Additionally, breast-cancer-associated ASCs interfere with the immunomodulatory function of natural killer cells [57]. These data suggest that the communication between obesity-associated ASCs, immune cells, and breast cancer cells may modify cellular compartments, leading to the co-evolution of cancer cells and their microenvironment [58].

## 4. Crosstalk between ASCs and Breast Cancer Cells

The crosstalk between ASCs and breast cancer cells can occur directly via cell–cell interaction and indirectly via the secretome released by the cells. Cancer cells secrete numerous chemotaxis signals [59], which recruit ASCs from local adipose tissues as well as MSCs from bone marrow into malignant tissue [60]. Cancer-educated ASCs/MSCs may differentiate into cancer-associated ASCs/MSCs or cancer-associated fibroblasts (CAFs) [61]. These ASCs/MSCs in turn promote breast cancer progression [62]. In particular, the ASC/MSC secretome, which is composed of a large number of secreted proteins, peptides and extracellular vesicles (EVs), is considered to be the main effector of their regenerative, tropic, trophic, angiogenic, and immunomodulatory functions.

Regarding the indirect manner, both breast cancer cells and ASCs/MSCs are potent in secreting a large number of soluble bioactive factors [61,63]. In particular, MSCs have the ability to migrate into malignant areas and stimulate cancer development by secreting a range of paracrine factors such as chemokines C-X-C ligand 1 (CXCL1), CXCL2, CXCL5, CXCL7, and CXCL12/stromal-cell-derived factor 1 (SDF1); cytokines such as IL6, IL8, and transforming growth factor β (TGFβ); and growth factors including epidermal growth factor (EGF), insulin-like growth factor 1 (IGF1), and vascular endothelial growth factor (VEGF) [58,61,64]. It has been shown that MSCs facilitate angiogenesis by paracrine secretion of angiogenic growth factors such as platelet-derived growth factor (PDGF) and VEGF [58,65]. Our own studies also demonstrate that ASCs isolated from subcutaneous as well as visceral adipose tissue released numerous cytokines, chemokines, and growth factors involved in inflammation, angiogenesis, and cell migration and proliferation, such as IL6, IL8, TNFα, CXCL1/2/3, CXCL5, monocyte chemotactic and activating factor (CCL2), and EGF [32,48,66].

Moreover, both breast cancer cells and MSC/ASC-derived EVs are essential for the crosstalk between MSCs/ASCs and breast cancer cells [63,67,68]. EVs are a heterogeneous group of membrane-bound vesicles released from cells by invagination and budding. They facilitate cell-to-cell interactions via contact with neighboring cells or internalization by recipient cells, which includes fusion with membrane and endocytosis [69]. According to the biogenesis, biophysical properties, and function, EVs can be classified into three main subtypes, namely exosomes (30–150 nm), microvesicles (MVs) (50–1000 nm), and apoptotic blebs (1000–5000 nm) [70]. Among these EVs, exosomes and MVs are of particular importance in cell–cell communication [70]. Both of them contain lipids, proteins, and genetic material, such as DNA, messenger RNA (mRNA), microRNA (miRNA), and long non-coding RNAs (lncRNA) that can be delivered to and reprogram the recipient cells [71]. Studies have shown that MSC/ASC-EVs exert both inhibitory and promoting effects in several situations and different stages of breast cancer. Through the transfer of various tumor-related factors, EVs promote proliferation, angiogenesis, metastasis, and drug resistance of malignant tumor cells [72,73], as shown in breast cancer cells stimulated with Her2-loaded EVs [72]. These data suggest that ASCs/MSCs may secrete molecules that act in concert with the secretome of breast cancer cells to remodel the microenvironment.

The direct crosstalk between ASCs and breast cancer cells is strictly dependent on their close contact that is established in the TME. Interestingly, tunneling nanotubes (TNTs) have emerged as a new important means of cell–cell communication. TNTs are thin membrane protrusions that connect cells over long distances allowing the exchange of various cellular components, including organelles, proteins, calcium ions, viruses, and bacteria [74]. Notably, TNTs are able to connect multiple cells forming functional cellular networks [75]. TNTs are therefore considered as novel bridges of intercellular communication in physiological and pathological cell processes [76]. Interestingly, MSCs have been shown to form TNTs and transfer mitochondria and other components to target cells [77,78,79]. This occurs under both physiological and pathological conditions, where cells are under stress, leading to changes in cellular energy metabolism and functions [76]. In this context, it is feasible to hypothesize that the protective role of ASCs/MSCs in breast cancer cell survival may be partially mediated through the formation of TNTs, in particular, when breast cancer cells are under stress from chemotherapy or radiotherapy.

Moreover, the activation of several signaling pathways requires direct cell–cell contact via their membrane-bound ligands and receptors, such as the canonical Notch signaling pathway [80]. Notch signaling is linked to the maintenance of breast cancer stem cells [81] and induction of epithelial-to-mesenchymal transition (EMT) resulting in an increase in migration and invasion of breast cancer cells [82]. In fact, direct co-culture of obese ASCs enhanced Notch signaling in ER^+^ breast cancer cells co-responsible for radiation resistance [83].

Finally, cell–cell fusion, a process that merges the lipid bilayers of two different cells, plays a crucial role during embryonic development as well as in tissue regeneration [84,85]. Studies also provide evidence that cell–cell fusion is closely related to cancer development and metastasis [86]. Although this highly regulated process is not yet fully understood, bone-marrow-derived cells were reported to be able to fuse to cancer cells, and the fused hybrids acquired more malignant characteristics and enhanced self-renewal ability [87]. In line with this observation, MSCs were reported to fuse with diverse malignant cells to promote proliferation and metastasis, including with lung cancer cells [88], liver cancer cells [89], and gastric cancer cells [90]. In particular, it was demonstrated that MSCs were fused with breast cancer cells and promoted their metastatic capacity [91]. Recently, is has been revealed that ASCs are able to fuse spontaneously with breast cancer cells, where breast cancer stem cell (CSC) markers CD44^+^CD24^−^/^low^EpCAM^+^ are enriched in this fused population [92]. These studies suggest cell fusion as a direct interaction between ASCs/MSCs and cancer cells. Further investigations are needed to explore the molecular mechanisms by which ASCs/MSCs and malignant cells are able to fuse and how this process promotes malignancy.

In sum, as illustrated in Figure 1, the crosstalk between ASCs/MSCs and breast cancer cells is multilateral and majorly mediated by indirect patterns such as the secretion of soluble bioactive factors and EVs released by ASCs as well as breast cancer cells. Although observed mainly in vitro, the interaction of breast cancer cells with ASCs/MSCs may be supported by direct cell–cell contacts including the formation of TNTs, binding of membrane-bound ligands to receptors, and cell–cell fusion. These communications may reshape the TME and fuel breast cancer progression and therapy resistance.

## 5. Mutual Interaction between ASCs/MSCs and Breast Cancer Cells

The communication between MSCs/ASCs and breast cancer cells has been an intensive research focus. The related studies are mostly performed using in vitro models to investigate the effects of ASCs/MSCs or their conditioned medium on proliferation, survival, migration, and invasion of breast cancer cell lines. Breast cancer cell lines are classified based on the status of three important cell surface receptors conventionally used for breast cancer subtyping, ER, PR, and HER2 [93]. Most studies used the following breast cancer cell lines: low metastatic breast cancer cell line BT474 (ER^+^, PR^+^, HER2^+^), MCF-7 (ER^+^, PR^+^, HER2^−^) and T47D (ER^+^, PR^+^, HER2^−^), metastatic breast cancer cell lines HCC1954 (ER^−^, PR^−^, HER2^+^, with wild type breast cancer gene 1 (BRCA1)), SKBR3 (ER^−^, PR^−^, HER2^+^, with wild type BRCA1), and MDA-MB-453 (ER^−^, PR^−^, HER2^+^, with wild type BRCA1), and highly metastatic breast cancer cell lines MDA-MB-231 (triple negative, with wild type BRCA1), MDA-MB-468 (triple negative, with wild type BRCA1), MDA-MB-436 (triple negative, mutated BRCA1), and SUM149 (triple negative, mutated BRCA1) [93]. Regarding ASCs/MSCs, while tumor adjacent cells [94] or “cancer-educated” MSCs [95] were recently used, most of the studies employed ASCs/MSCs isolated from non-breast sources, including abdominal adipose tissue, bone marrow, and peripheral blood [32,58,96]. It is well-known that ASCs/MSCs from different tissues and organs have distinct transcriptomic, biochemical, and secretory profiles, as well as biologic functions in tissue-specific homeostasis, immune modulation, and vasculogenesis/angiogenesis [97]. This, together with other diversities, such as different BMI, varied donor age, variable ASC/MSC passages, and individual experiment settings, often leads to inconclusive results with breast-cancer-supportive and -suppressive functions [58,98], which may not reflect the situation in vivo in breast cancer tissue.

### 5.1. ASCs/MSCs Influence Breast Cancer and Related Molecular Mechanisms

Much attention has been paid to elucidating how ASCs/MSCs impact breast cancer cells as well as their TME (Table 1). Although their exact roles are not yet completely understood, ASCs/MSCs are described as both pro- or anti-tumorigenic, depending on the type and source of ASCs/MSCs, the use of breast cancer cell lines, and the in vitro or in vivo models. The studies concerning the anti-tumorigenic impact of MSCs are limited. MSCs have been reported to exert their negative impact on breast cancer by impairing angiogenesis via secretion of exosomes [99], reducing migration and invasion via the release of tissue inhibitor of metalloproteinase (TIMPs) [100], and decreasing breast tumor growth via down regulation of the STAT3 signaling pathway [101]. Nevertheless, the majority of studies report pro-tumorigenic effects of ASCs/MSCs on breast cancer cells, which are multilayered, as depicted in Table 1. The related molecular mechanisms are discussed in detail.

#### 5.1.1. Promoting Proliferation and Survival

ASCs/MSCs secrete various growth factors including IGF1 and EGF, and numerous cytokines such as leptin, IL6, adipsin, and TNFα, which stimulate proliferation and survival of breast cancer cells [67,102,103,116,120,121]. In particular, obesity is associated with an increase of IL6 in the circulation, reinforcing systemic inflammation [128]. Interestingly, increased IL6 was correlated with poor prognosis, progression, and migration of ER-positive breast cancer [129]. IL6 was shown in vitro to activate the signal transducer and activator of transcription 3 (STAT3)/protein kinase B (AKT)/mitogen-activated protein kinase (MAPK) pathways, triggering proliferation of both triple negative and triple positive breast cancer cell lines [103,104,130]. TNFα, another important inflammatory cytokine released by adipose stromal cells, including ASCs/MSCs, was increased in the TME of patients with obesity, causing adipose tissue inflammation and inhibiting apoptosis of TNBC cells [131,132]. This elevated level of TNFα in individuals with obesity might establish a positive feedback loop, since TNFα was shown to activate ASCs/MSCs and stimulate their secretion of multiple cytokines, such as chemokine (C-C motif) ligand 5 (CCL5), CXCL1, CXCL2, and CXCL5, which significantly enhanced tumor growth and metastasis [115,116]. Morbid obesity, defined as BMI equal to or greater than 40 [133], is associated with hyperleptinemia and increased leptin impairs the negative feedback mechanism between the adipose tissue and neurons in the hypothalamus [134,135]. In support of this notion, ASCs isolated from obese individuals secreted significantly higher levels of leptin that stimulated the proliferation of low and high malignant breast cancer cells [121,122]. This is linked to leptin receptor activation, which triggers multiple pathways, such as Janus kinase (JAK) and MAPK, with the expression of downstream target genes involved in cell cycle progression and proliferation, including cyclin D1 (*CCND1)*, *VEGF*, and proto-oncogene C-Fos (*FOS),* transcription factor AP-1 subunit jun (*JUN),* and transcription factor AP-1 subunit JunB (*JUNB)* [136]. Adipsin is another adipokine upregulated in ASCs derived from obese patients, which stimulates the cell surface receptor complement C3a receptor 1 (C3aR) and the cleavage of factor B, leading to proliferation of breast cancer cells [120]. Moreover, obesity is associated with increased levels of circulating IGF1, also secreted by ASCs/MSCs [137]. Breast cancer cells express IGF1 receptors, and binding of IGF1 activated phosphoinositide 3-kinase (PI3K) and MAPK pathways, promoting cancer cell proliferation [138,139,140]. Similarly, serum levels of hepatocyte growth factor (HGF) were elevated, which is secreted by stromal cells, including ASCs, during obesity [123], and its receptor, tyrosine-protein kinase Met (c-Met), is expressed on breast cancer cells [126]. Thus, increased expression of HGF promoted c-Met-induced cell proliferation and subsequent progression of breast cancer [141,142]. In addition, ASCs/MSCs are capable of modulating the metabolism of breast cancer cells by stimulating, for example, the upregulation of cluster of differentiation 36 (CD36), a fatty acid receptor, leading to an increased proliferation rate [113], or the upregulation of S100 calcium-binding protein A7, involved in cell cycle regulation [114]. Moreover, numerous studies demonstrate that cancer cells are able to educate ASCs/MSCs, resulting in an altered gene and protein expression [143]. The interaction with cancer cells increased the release of tumor-promoting cytokines such as CCLs, CXCLs, SDF, and EGF [105,117]; angiogenesis factors including VEGF, angiopoietins, EGF, galectin-1, IGF1, and keratinocyte growth factor (KGF) [102,115,118,144]; and EMT inducers such as TGFβ, platelet-derived growth factor D (PDGF-D), and stem cell factor (SCF) [106,118,145]. These data strongly suggest that ASCs/MSCs promote proliferation and survival of breast cancer cells by releasing diverse bioactive factors activating various signaling pathways.

#### 5.1.2. Stimulating Tumor Angiogenesis

ASCs/MSCs are regarded as an important cell type influencing vascular repair mechanisms [146] and as inducers of neovascularization [147]. The proposed models by which ASCs/MSCs facilitate these functions are diverse, including direct cell–cell contact [148], differentiation into endothelial cells (ECs) [147], and paracrine signaling [149]. The molecular mechanisms depend on the release of angiogenic factors such as angiopoietin 1 (Ang1), Ang2, VEGF, TGFβ, SCF, and von-Willebrand factor (vWF); lipids including fatty acids, phospholipids, ceramide, and sphingolipids; microRNAs such as miR-181b-5p, miR-494, miR-125a, and miR-210; and signaling molecules including wingless/integrated 3a (Wnt3a), Wnt4, and matrix metalloprotease (MMP) inducer in ECM [146,147,149]. Obesity influences the direct cell–cell interaction, and the paracrine signaling and differentiation ability of ASCs/MSCs [10,50,150]. Consistently, it was shown that obese ASCs/MSCs had deficient angiogenic properties [151,152], and were not able to promote VEGF expression and tube formation of injured human umbilical vein endothelial cells (HUVECs) [153]. Moreover, EVs secreted by obese ASCs lost their capacity to stimulate angiogenesis in endothelial cells, possibly by a significantly reduced expression of miR-126, leading to an upregulation of sprouty-related EVH1 domain containing 1 (Spred1) and an inhibition of extracellular-signal regulated kinase (ERK1/2) essential for endothelial cell angiogenesis [154]. A recent high-throughput sequencing analysis presents a more complex picture [155]. This study analyzed secreted EVs from obese and lean ASCs, and 83 miRNAs were found to be significantly deregulated in obese ASCs, with significant implications in angiogenesis [155]. Clinical data provide further evidence that obesity is associated with resistance to anti-VEGF therapies, enlarged tumor size and increased vascularization in breast cancer patients [156,157,158]. This could be explained on several levels, including increased IL6 and other inflammatory cytokines released by ASCs [66], macrophages, and adipocytes, which were shown to trigger resistance toward anti-VEGF therapy [156]. Moreover, an increased secretion of IL1β, which is also significantly elevated in obese ASCs [159], was identified to trigger an NLR family CARD-domain-containing protein 4 (NLRC4)-dependent upregulation of angiopoetin-like 4 (ANGPTL4), which is a known angiogenic factor in the TME of breast cancer [158]. Its genetic knockout prevented obesity-induced enhanced angiogenesis in mice [158]. Leptin is described as another major driver in the context of obesity-induced angiogenesis [157], which stimulates the expression of VEGF by activating the hypoxia-inducible factor 1-alpha HIF1α and NFĸB pathways [160]. Interestingly, inhibition of leptin signaling also decreased the vascular endothelial growth factor receptor 2 (VEGFR-2) expression levels in endothelial cells and breast cancer cells [161]. Collectively, ASCs/MSCs display a pro-angiogenic phenotype in the TME through diverse signaling.

#### 5.1.3. Escape of Immune Response

To escape anti-tumor immunity, cancer cells exploit cell-intrinsic pathways associated with resistance to immune cell-mediated attack and avoid recognition by anti-tumor immune cells [162,163,164]. Cancer cells may also enhance immunosuppression of the TME by regulating the expression or secretion of immunosuppressive molecules, including cytokines and chemokines. On the one hand, this intercellular communication network effectively inhibits immune effector cells, including T-cells, natural killer (NK) cells, and dendritic cells (DCs). On the other hand, it promotes the functions and/or the recruitment of immunosuppressive cells such as regulatory T-cells (Tregs) and tumor-associated macrophages [165,166]. Cancer cell escape from the immune response is mediated mainly by paracrine and autocrine stimulation in the TME by a variety of growth factors and cytokines, including TGFβ, basic fibroblast growth factor (bFGF), VEGF, PDGF, and ILs [167,168].

ASCs/MSCs in the TME are well-known for their exceptional immunomodulatory capacity. They have a low expression of major histocompatibility complex (MHC) class I, and expression of class II MHC molecules is completely absent [169], which helps these cells evade immune recognition. Moreover, ASCs/MSCs are capable of modulating the immune response by suppressing lymphocytes proliferation, inhibiting differentiation of monocyte-derived immature DCs, and reducing the cytotoxic activity of NK cells [170,171]. Their functions are supported both by direct cell–cell interaction and by paracrine signaling through the release of multiple cytokines and other soluble factors [171,172]. Intriguingly, cancer cells were shown to exploit the immunomodulatory capacity of ASCs/MSCs. The supernatant of ASCs, which were isolated from breast cancer tissue, was reported to upregulate a panel of anti-inflammatory cytokines, such as IL4, IL10, CCR4, CD25, and TGFβ in peripheral blood lymphocytes (PBLs) and to increase the number of Tregs, which could establish an anti-inflammatory reaction in the TME [110]. It was also reported that breastcancer-educated MSCs enhanced the proliferation of PBLs by higher secretion of TGFβ, prostaglandin (PGE2), indoleamine 2,3-dioxygenase (IDO), and VEGF [111]. In line with these results, the co-culture of cancer associated ASCs with T-cells expanded the CD25^+^FOXP3^+^CD73^+^CD39^+^Treg population and increased the release of immune suppressive cytokines IL10, IL17, and TGFβ [173]. In addition, indirect co-culture of ASCs with activated PBLs reduced the number of killer cell lectin-like receptor K1 (NKG2D^+^) and CD69^+^ NK cells [57]. Interestingly, reduced proliferation of B-cells and TNFα^+^/IL10^+^ cells was observed only in direct co-culture but not in indirect co-culture experiments [112], suggesting the importance of direct cell–cell contact. These data demonstrate that ASCs/MSCs greatly contribute to the immune escape by affecting the proliferation and function of diverse immune cells such as PBLs and T- and B-cells in the TME.

Obesity is a key factor influencing the immunomodulatory capacity of ASCs/MSCs, mainly by altering the cytokine secretion profile with a loss of anti-oxidant molecules such as glutamate-cysteine ligase (GCL), peroxiredoxin-5 (Prdx5) and Prdx6, as well as a loss of functions of the tissue development regulators including Ang, Angptl4, follistatin-related protein 3 (Fstl3), and placental growth factor (PLGF) [174]. In contrast, obesity triggers the secretion of cytokines involved in osteoporosis, negative vessel remodeling, and inflammation with increased leukemia inhibitory factor (LIF), IL1β, CCL2, leptin, interferon gamma (IFNγ), IL6, and TNFα [66,121,159,174,175]. This switch in the secretome of ASCs/MSCs contributes to adipose tissue inflammation in patients with morbid obesity [10], as highlighted by multiple investigations [175,176,177,178]. The in vitro co-culture of obese ASCs with mononuclear cells enhanced monocyte and Th17 activation [178] and programmed cell death ligand 1 (PD-L1) expression, decreased the cytokine secretion in Th1 cells, and reduced the cytolytic activity in Th1 cells dependent on the elevated IFNγ release of ASCs [175]. It was also shown that obese ASCs lost their ability to regulate the polarization of M1/M2 macrophages in vitro and in vivo [176], which was associated with a four-fold higher concentration of TNFα in the supernatant of obese ASCs [176]. Accordingly, it was reported that co-culture with obese ASCs promoted a pro-inflammatory phenotype in murine macrophages and microglial cells through increased expression of genes involved in inflammation, altered nitric oxide activity, and impaired phagocytosis [177]. Strikingly, Benaige et al. found that obese ASCs induced a different phenotypic switch in macrophages with pro- and anti-inflammatory features, leading to a tumor-associated macrophage phenotype [179]. These authors also reported that co-cultured macrophages secreted survivin, stimulating the progression of cancer cells [179]. Beyond ASCs/MSCs’ secretome, ASC-orchestrated ECM regulation is crucial in restricting access of immune cells to cancer, by generating a physical barrier to tumor infiltration, inhibition of cytotoxic response, and the drug diffusion [180,181]. In conclusion, the immunosuppressive features of cancer-educated ASCs/MSCs, particularly obese ASCs/MSCs, may greatly contribute to the immune evasion of breast cancer cells.

#### 5.1.4. Inducing EMT, Migration and Invasion

EMT is the trans-differentiation process that causes epithelial cells to lose their epithelial characteristics, such as cell junctions and apical-basal polarity, and acquire mesenchymal features, promoting cell motility and invasion [182]. ASCs/MSCs release cytokines and growth factors such as TGFβ, EGF, PDGF, and HGF, which trigger EMT [183,184]. The main signaling pathways that induce EMT include the TGFβ, Wnt/β-catenin, and Notch pathways [185]. All these pathways converge on the expression and activation of the transcriptional factors such as Snail, Slug, Twist-related protein (TWIST), forkhead box C1 (FOXC1), FOXC2, and zinc finger E-box binding homebox 1/2 (ZEB1/2). These transcription factors suppress the expression of adherens junction and integrin proteins, which causes tumor cells to lose their polarity and dissociate from adjacent cells and the basal membrane [186,187]. While Snail, Slug, and ZEB2 are able to directly repress the E-cadherin promoter, TWIST1, FOXC2, and ZEB1 possess an indirect molecular mechanism, which disrupts cell polarity and gives rise to the mesenchymal phenotype [186,188]. In particular, TWIST1 is able to promote transformation of normal mammary epithelial cells into mesenchymal-like cells with increased expression of vimentin, N-cadherin, and fibronectin [189]. Moreover, ZEB1 also plays an important role in EMT regulation in breast cancer cells [190], dramatically increasing the metastatic rate, plasticity, and therapy resistance of breast cancer [191]. 

MSCs/ASCs are potent in promoting the EMT process of breast cancer cells and foster their migration and invasion through various pathways. S1007A is a protein that has been shown to be strongly upregulated in breast cancer cells treated with conditioned medium from ASCs, and this upregulation was associated with markedly increased migration [114], possibly by inducing EMT as reported in cervical cancer cells [192]. MSCs/ASCs are capable of reshaping the TME by secreting lots of MMPs, including MMP1, MMP2, MMP3, MMP8, MMP9, MMP10, and MMP13 [193,194,195], which degrade the tumor-associated ECM. The proteolytic degradation of ECM generates bioactive matrikines and releases matrix-bound VEGF, supporting the growth, migration, and metastasis of cancer cells [196]. The engulfment of MSCs by breast cancer cells is another cellular mechanism by which cancer cells promote their migration, invasion, metastasis, and self-renewal capacity [107]. The process increased the gene expression of oncogenic factors such as cellular tumor antigen p53 (*p53*), *WNT5A*, Myc proto-oncogene protein c (*c-MYC*), *TGFβ,* and cell-membrane-associated genes including macrophage scavenger receptor types I and II (*MSR1*), engulfment and cell motility protein 1 (*ELMO1*), interleukin 1 receptor-like 2 (*IL1RL2*), zona pellucida-like domain-containing protein 1 (*ZPLD1*), and signal-regulatory protein beta-1 (*SIRPB1*) [107]. In fact, low expression of *WNT5A* and *MSR1* was linked to reduced metastasis and longer cancer-free survival of breast cancer patients [107,197]. In addition, it was reported that breast cancer cells treated with ASC supernatant upregulated their fatty acid receptor CD36, which is associated with migration and invasion [113]. 

Interestingly, various reports showed a reinforced effect of obese ASCs/MSCs on the malignancy of breast cancers. This could be attributed to several aspects. First, this enhanced effect could be induced by an increased secretion of cytokines, as shown for IGF1 [123], which stimulated the invasiveness of breast cancer cells by activating its downstream targets ERK, serine/threonine-protein kinase mTOR, and STAT3 [137], and for leptin, with the expression and activation of its various effector genes [83,124]. Second, obese ASCs were described to induce the invasion of breast cancer cells more efficiently compared to lean ASCs by direct cell–cell contact, helping the generation of traction forces within the TME and releasing various MMPs [125]. Third, obesity is associated with an enhanced de-differentiation of ASCs into CAFs, which changes significant parts of their secretome toward a cancer-supporting phenotype [126,198]. The changed secretome might explain the changes observed in constitutively active ER^+^ breast cancer cell lines upon co-culture with obese ASCs [127]. In conclusion, ASCs/MSCs have the ability to support breast cancer cell migration through multiple pathways, in particular through promoting EMT and reshaping the ECM, which is abused by cancer cells and fueled by morbid obesity on various molecular levels to increase cancer migration, invasion, and metastasis.

#### 5.1.5. Raising Cancer-Associated Stem Cells

The term cancer stem cell (CSC) characterizes a subpopulation of cancer cells with an intrinsic self-renewal and tumorigenic capacity, mirrored by their significant role in tumor development, therapy resistance, relapse, and metastasis [199]. The pathways responsible for establishing a CSC phenotype are diverse and differ among cancer entities [200]. In breast cancer cells, the most important pathways for this process are STAT, Hedgehog, protein kinase C (PKC), MAPK, Notch, and Hippo, with hundreds of downstream genes responsible for enhanced stemness [201]. Obesity increases systemic levels as well as cellular secretion of many cytokines and adipokines including leptin, IL6, TNFα, IGF1, fatty acid binding protein 4 (FABP4), and resistin, which are all involved in regulating the pathways associated with the development of CSCs in breast cancer tissue [202]. In accordance with this, Sabol et al. showed that patient-derived xenograft (PDX) tumors co-cultured with obese ASCs increased the formation of metastases and the number of CD44^+^CD24^−^ breast cancer stem cells in a severely immunodeficient (SCID) mouse model [124]. Remarkably, the stable knockdown of leptin in obese ASCs led to a significant reduction in circulating CSCs [124], suggesting leptin as a key factor to induce CSCs by obese ASCs. Furthermore, leptin was reported to stimulate the secretion of TGFβ, which leads to the activation of SMAD family member 2 (Smad2), Smad3, and Smad4 transcription factors, the repression of cadherin 1 (CDH1) coding for E-cadherin, and an increased CSC phenotype in breast cancer cells [203]. Moreover, leptin-induced TGFβ was shown to trigger de-differentiation of stromal cells in the TME, including fibroblasts, ASCs, and MSCs, toward a cancer-associated phenotype [204], which altered their cytokine secretion pattern and in turn increased TGFβ secretion [205]. Resistin, another adipokine with an increased secretion in obese adipose tissue [206], was highly associated with the transcription of pluripotency genes such as aldehyde dehydrogenase 1 family member A1 (*ALDH1A1*), *ITGA4*, protein lin-28 homolog B (*LIN28B*), smoothened homolog (*SMO),* and sirtuin 1 (*SIRT1*) in low malignant breast cancer cells and non-carcinogenic breast epithelial cells [207]. Finally, the obese adipose tissue is characterized by systemic and local chronic inflammation with a highly increased level of circulated IL6 [208], which is also secreted by ASCs in obese adipose tissue [66]. This elevated IL6 level was associated with the activation of the JAK2/STAT3 signaling cascade and increased levels of SRY-Box transcription factor 2 (*SOX2*), Nanog homeobox (*Nanog*), *ALDH1A1,* and ATP-binding cassette subfamily G member 2 (*ABCG2*) genes in breast cancer cells in vitro and in vivo [119], which was completely prevented by blocking the cellular IL6 signaling [119]. These data strongly suggest that ASCs/MSCs have the potential to stimulate breast cancer stemness, which is significantly enhanced by factors associated with obesity.

#### 5.1.6. Facilitating Therapy Resistance

ASCs/MSCs reshape the TME, promote EMT, and support the generation of CSCs, which are associated with radio- and chemotherapy resistance [199,209]. The co-culture of ER^+^ breast cancer cells with obese ASCs activated many pathways such as leptin, IL6, Notch, and jagged canonical Notch ligand 2 (JAG2), which mediated radiation resistance in ER^+^ breast cancer cells [83]. Blocking either leptin or IL6 from the culture medium prevented this radiotherapy resistance [83]. Beside the secretion of cytokines, the direct co-culture of breast cancer cells with ASCs/MSCs activated TGFβ/Smad, PI3K/AKT, and MAPK signaling, leading to the induction of chemotherapy resistance after 72 h even in a time frame before EMT occurred [32,108]. Interestingly, an in vivo experiment showed that specific depletion of ASCs within the TME by selective peptide targeting D-CAN, consisting of ASC binding domain and a pro-apoptotic domain, resulted in decreased cisplatin and paclitaxel resistance in a human breast cancer xenograft model [109]. In support of this observation, another study reported that ASC removal increased the efficiency of cisplatin and suppressed obesity-induced EMT in obese mice with prostate cancer [210]. In addition, ASCs/MSCs were also reported to increase the chemotherapy resistance of other cancer entities, including colorectal [211], ovarian [212,213,214], lung [215], squamous cell carcinoma [216], and acute myeloid leukemia [217], by modulating PDGF-BB [213], X-linked inhibitor of apoptosis (XIAP) [218], Notch [217], STAT3 [101], Hedgehog [214], and p53 signaling [211]. These data highlight the general role of ASCs/MSCs in rendering cancer cells resistant to radio- and chemotherapy by activating diverse signaling, in particular, by generating EMT and CSCs, and by reshaping the TME.

In sum, ASCs/MSCs, particularly in the obese state, promote the development of breast cancer by facilitating cell proliferation and survival, EMT, migration and invasion, angiogenesis, CSC formation, immune escape, and therapy resistance (Figure 2). While most reports emphasize supportive effects for breast cancer cells, ASCs/MSCs have also been shown to have an anti-tumorigenic function [101,219]. Secreted factors of human MSCs isolated from umbilical cords were shown to suppress tumor progression and increase radiosensitivity through downregulating intra-tumoral STAT3 signaling in a xenograft mouse model and in breast cancer cell lines [101]. Additionally, ASC supernatant was reported to be able to induce cell death with increased caspase-3/7 activity, whereas this was associated with the augmentation of stemness in breast cancer cells [219]. Further studies are needed to clarify the relationship between ASC/MSC and breast cancer cells.

### 5.2. Breast Cancer Cells Educate ASCs/MSCs and Related Molecular Mechanisms

While ASCs/MSCs influence breast cancer cells, numerous investigations show that breast cancer as well as its TME “educate” their surrounding cells, including fibroblasts (FBs) and ASCs/MSCs, toward pro-tumorigenic phenotypes [220]. The most precisely characterized cells are FBs. As depicted in Table 2, multiple cancer-associated fibroblast (CAF) phenotypes have been identified during the last decade as key components of the TME with implications in tumor growth, therapy resistance, metastasis, ECM remodeling, and immune tolerance [34,205,221]. Interestingly, ASCs/MSCs are morphologically indistinguishable from fibroblasts. These two cell types share many common features including their surface marker composition, proliferation pattern, differentiation capacity, immunomodulation property, and, to some extent, even their gene expression profiles [222,223]. The major difference between these two cell types seems to be their methylation profile. While the general methylation patterns of MCSs are maintained in long-term culture and aging [224], the methylation of fibroblasts seems to decrease with aging or prolonged culture [225]. Indeed, ASCs/MSCs have been proposed to be immature FBs and one of the sources for FBs [222]. 

Recently, increasing evidence highlights that ASCs/MSCs are educated and de-differentiated by cancer cells and the TME, fueling malignancy and therapy resistance [226,227]. Cancer-cell-secreted factors and direct cancer cell–ASC/MSC contacts induce a pro-tumorigenic population of ASCs/MSCs, named cancer-associated MSCs (CA-MSCs) [143]. CA-MSCs have the ability to differentiate into multiple cell lineages, such as fibroblasts and adipocytes, suggesting that MSCs may play a key role in the generation of most stromal components of the TME. A number of reports have demonstrated that CA-MSCs differentiate into CAFs and cancer-associated adipocytes (CAAs) in the presence of malignant cells [105,228]. While the exact mechanisms underlying the de-differentiation of CA-MSC are not yet clear, this switch resulted in a highly secretory phenotype with increased secretion of bone morphogenetic protein (BMP2), BMP4, and IL6 [229]. In line with this observation, there was evidence suggesting that cancer-released TGFβ was able to activate the Smad signaling pathway in MSCs, which drove differentiation into a cancer-associated phenotype [230]. In other tumor entities, including lymphomas [115], lung [215], and gastric cancer [231,232], IL6, IL8, IL17, IL23, and TNFα secreted by monocytes, macrophages, neutrophils, and non-MSC stromal cells were shown to be capable of promoting malignant transition of ASCs/MSCs, which was associated with significantly increased metastatic rates and tumor growth [115,215,227,231,232]. Other molecular mechanisms proposed for CAF activation include Notch/Eph-ephrin signaling, ECM composition in the TME, DNA damage, physiological stress, inflammatory stimuli, RTK ligands, and TGFβ-mediated signaling [34]. Moreover, the primary cilium, a sensory organelle with an exceptionally high receptor density [233], was shown to play a critical role in the de-differentiation process of adipose progenitors toward a CAF phenotype by mediating TGFβ signaling [234]. All these studies suggest that diverse pathways are responsible for the activation of CAFs, and the TME is likely the major player in triggering the de-differentiation of ASCs/MSCs into different CAFs, depending on their cellular context and the tumor entity.

**Table 2 cancers-14-03908-t002:** Subtypes of CAFs in different cancer entities.

Fibroblast/CAF Source	Study Design	Functions and Molecular Mechanisms	Ref.
** Impact of cancer cells on the fibroblast phenotype **			
FBs and CAFs isolated from surgical explantation and human BM-MSCs obtained from AOU Meyer Hospital (Florence)	Co-culture experiments with FBs, CAFs, and BM-MSCs with PC3, DU145, and LNCaP prostate cancer cell lines in vitro	Prostate cancer cells secreted TGFβ1 and recruited BM-MSCs into the TME. This in turn led to an elevated secretion of TGFβ1 in cancer-educated BM-MSCs. Blocking TGFβ1 reduced the recruitment of BM-MSCs into the tumor as well as their trans-differentiation.	[230]
Pancreatic ductal adenocarcinoma (PDAC) tissue	Single-cell RNA sequencing to characterize CAF subpopulations in PDAC	The analysis revealed intertumoral heterogeneity between CAFs, ductal cancer cells, and immune cells in extremely dense and loose types of PDACs. A highly metabolic active subtype (meCAFs) was identified. Patients with abundant meCAFs had a significantly increased risk for metastasis and poor prognosis. These patients, however, showed a highly increased response to immunotherapy.	[235]
Murine normal pancreatic and cancer tissue	Single-cell RNA sequencing to characterize CAF subpopulations (normal vs. pancreatic cancer tissue)	The analysis revealed a landscape of CAFs in pancreatic cancer during in vivo tumor development. The LRRC15^+^ CAF lineage was shown to be TGFβ-dependent and correlated with a poor patient outcome treated with immunotherapy in multiple solid tumor entities.	[236]
Human and murine PDAC resection specimens and normal pancreas tissue	Single-cell RNA sequencing to characterize CAF subpopulations (human and murine)	The analysis from neoplastic and TME of human and mouse PDAC tumors displayed already described myCAFs and iCAFs with distinct gene expression profiles. It further revealed a novel subtype that expressed MHC class II and CD74 called “antigen-presenting CAFS (apCAFs)”. These cells activated antigen-specific CD4^+^ T cells. These immunomodulatory CAFs were likely associated with a reduced immune response of PDAC tumors.	[237]
Human breast and BC tissue	Single-cell RNA sequencing to characterize CAF subpopulations (normal vs. BC tissue)	The analysis identified different CAF subpopulations in BC tissue. CAF-S1 (CD29, FAP, α-SMA, PDGFRβ, FSP1, and CXCL12) was analyzed in detail. These cells induced an immunosuppressive TME by retaining CD4^+^CD25^+^ T cells through the signaling of OX40L, PD-L2, and JAM2, and increased CD25^+^FOXP3^+^ T lymphocytes, and B7H3, DPP4, and CD73 signaling.	[238]
Human BC tissue and metastatic lymph nodes tissue (LN)	Single-cell RNA sequencing to characterize CAF subpopulations (BC and LN tissue) and co-culture experiments with MCF7, MDA-MB-231, and T47D	The analysis identified four CAF subpopulations in LN. Two had a myCAF gene expression pattern, CAF-S1 and CAF-S4, accumulated in LN and correlated with cancer cell invasion. CAF-S1 stimulated cancer cell migration by stimulating EMT, through CXCL12 and TGFβ signaling. CAF-S4 induced cancer cell invasion through Notch signaling. Patients with a high ratio of CAF-S4 cells were prone to develop late distant metastases.	[239]
Murine BC tissue and normal mammary fat pad tissue	Single-cell RNA sequencing to characterize CAF subpopulations (BC compared to pancreatic cancer tissue)	The study identified six CAF subpopulations in a triple-negative syngeneic breast cancer mouse model. Among these six subpopulations, myCAFs, iCAFs, and apCAFs were found to exist in BC cancers and PDAC. The subtype expressing MHC class II proteins similar to apCAFs were also found in normal breast/pancreas tissues, indicating that this specific subtype is not TME induced. The comparison to a pancreatic tumor model suggested that similar phenotypes exist in both cancer entities without a TME-specific subtype.	[240]
Murine and human BC tissue and normal mammary fat pad tissue	Single-cell RNA sequencing to characterize CAF subpopulations (murine, human BC tissue vs. normal mammary fat pad tissue) and co-culture experiments with human MDA-MB-231 as well as murine 4T1 and EO771.	A negative selection strategy was used to analyze 768 single-cell RNA sequencing transcriptome data of mesenchymal cells in a BC mouse model. In this approach, three distinct CAF subpopulations were defined. These populations were named “vascular”-CAFs, “matrix”-CAFs and “development”-CAFs. The found gene signatures were further verified on the transcriptional and protein levels in various experimental cancers. Human tumors and every CAF gene profile were correlated with distinctive molecular functions.	[241]
Normal breast, BC tissue samples, and metastatic lymph nodes obtained from surgery	Comparison of multiple genome transcriptomic RNA sequencings	These approaches revealed that most of the described cancer hallmark signaling pathways were significantly upregulated in triple-negative breast cancer with a highly enriched CAF population. BGN, a soluble secreted protein, was upregulated in CAFs compared to normal cancer-adjacent fibroblasts (NAFs). The expression was negatively associated with CD8^+^ T cells and poor prognostic outcomes.	[242]
Human primary bladder tumor tissues and adjacent normal mucosae tissues	Single-cell RNA sequencing to characterize CAF subpopulations (bladder cancer tissue vs. normal mucosae tissue)	iCAFs were identified as poor prognostic marker with potent pro-proliferation capacities, and their immunoregulatory function in the TME of bladder cancer was further deciphered. The LAMP3^+^ dendritic cell subgroup might be able to recruit regulatory T cells, which could be a step toward an immunosuppressive TME.	[243]

Abbreviations: FB, fibroblast; BC, breast cancer; CAF, cancer-associated fibroblast; apCAF, antigen presenting cancer-associated fibroblast; meCAF, metabolic active subtype cancer-associated fibroblast; iCAF, inflammatory cancer-associated fibroblast; myCAF, myofibroblast cancer-associated fibroblast; BM-MSCs, bone-marrow-derived mesenchymal stromal/stem cells; PDAC, pancreatic ductal adenocarcinoma; TME, tumor microenvironment; CD29, cluster of differentiation 29; α-SMA, smooth muscle actin; PDGFRβ, platelet-derived growth factor receptor beta; IL6, interleukin 6; BGN, biglycan; LAMP3, lysosomal-associated membrane protein 3; FSP1, fibroblast-specific protein-1; CXCL1, C-X-C motif chemokine ligand 1; JAM2, junctional adhesion molecule 2; DPP4, dipeptidyl peptidase 4; B7H3/CD273, cluster of differentiation 273.

#### 5.2.1. CAF Subtypes and De-Differentiation of MSCs/ASCs into CAFs

Single-cell RNA sequencing is a tool to classify multiple different subtypes of CAFs, with a specific gene signature for many tumor entities [235,244]. In general, the following three major phenotypes have been described: myCAFs (myofibroblast-like CAFs) with a high expression of smooth muscle actin (αSMA), TGFβ signaling and the capacity to remodel the ECM [235]; iCAFs (inflammatory CAFs), defined by an increased secretion of inflammatory cytokines, chemokines, and the complement complex [236]; apCAFs (antigen-presenting), featured as a cell type able to induce T-cell receptor ligation in CD4^+^ T cells in an antigen-dependent manner and express CD74- and MHC-class-II-related genes [237].

Strikingly, a recent study was able to recapitulate the de-differentiation of human ASCs into similar phenotypes in a pancreatic cancer stromal-rich xenograft model [245]. ASCs were shown to de-differentiate into three major subpopulations, myCAF, iCAF, and apCAF [245], demonstrating that ASCs/MSCs are also susceptible to gaining malignant phenotypes in an in vivo mouse model [245]. Furthermore, a computational analysis of single-cell gene expression from pan-cancer biopsies could recapitulate the transition of ASCs into CAFs expressing COL11A1 [246]. This phenotypical switch appears to be dependent on the interaction between the TME and ASCs. In vitro experiments showed that the direct co-culture of ASCs with human pancreatic cancer cells led to the gene expression profile of myCAFs, whereas the indirect co-culture with the same cells induced an iCAF gene expression pattern [247]. In line with this observation, the direct co-culture with low malignant breast cancer cells and TNBCs was shown to stimulate the transition toward a myCAF-like phenotype driven by TGFβ/Smad3 signaling [94].

Interestingly, obesity seems to fuel this de-differentiation process. It was reported that obese ASCs expressed significantly higher levels of myCAF-associated genes, including actin alpha 2 (*ACTA2*), fatty-acid-binding protein 1 (*FAP1*), fibroblast-specific protein-1 (*FSP1*), and chondroitin sulfate proteoglycan (*NG2*), compared to lean control ASCs [126]. These cells also displayed an increased secretion of pro-tumorigenic cytokines, such as TARC (CCL17), CCL5, IL24, and IL6 [126]. This could be of clinical relevance, since breast cancers from obese women have an elevated incidence of desmoplasia, and these desmoplastic tumors are described as highly fibrillar collagen enriched with an increased number of CAFs [40,248], which can be further deciphered into different CAF subtypes with individual roles inside the TME [249,250]. In fact, subtypes of cancer associated ASCs/CAFs have been identified in breast cancer, as demonstrated in Table 3.

#### 5.2.2. De-Differentiation of MSCs/ASCs into myCAFs Remodeling ECM

The myCAF subpopulation is characterized by an upregulation of genes involved in smooth muscle contraction, focal adhesion, ECM organization, and collagen formation [237]. Additionally, myCAFs are associated with gene expression of *ACTA2*, transgelin (*TAGLN*), myosin light chain 9 *(MYL9*), tropomyosin 1/2 (*TPM1/2*), *FAP*, *FSP1,* and platelet-derived growth factor receptor beta (*PDGFRβ*) [253]. This subpopulation was shown to be essential for fibrosis in the TME, causing increased density and stiffness [254]. This highly fibrotic ECM decreased T-cell infiltration, and was associated with a hypoxia-induced metabolic switch, further suppressing the immune response in the TME [254]. As a consequence, myCAFs stimulated tumor proliferation, migration, and invasion [253]. In the TME of breast cancer, myCAFs were shown to promote an immunosuppressive environment by attracting and retaining CD4^+^CD25^+^ T-cells through the ligands tumor necrosis factor receptor superfamily member 4 (OX40L), PD-L2, and the adhesion molecule junctional adhesion molecule B (JAM2). Additionally, they were able to increase the number of CD25^+^FOXP3^+^ T-cells through dipeptidyl peptidase 4 (*DPP4*), *CD73,* and *B7H3* (cluster of differentiation 276) signaling [238]. As a direct effect on breast cancer, these cells were reported to trigger EMT by activating the CXCL12 and TGFβ pathways in breast cancer cells [239]. This significantly increased EMT process was associated with enhanced tumor migration and lymph-node metastasis [253]. In addition, a single-cell transcriptomic analysis comparing tumor-derived FBs and normal tissue-resident FBs revealed that about 79% of CAFs exhibited a myCAF phenotype with high gene levels of αSMA in 4T1 murine breast tumors [240]. Another study further defined these αSMA-positive cells into matrix CAFs (mCAFs), with a specific function in matrix remodeling [241]. Interestingly, a high prevalence of mCAFs was observed at the invasive front of cancers and a low abundance in the cancer core [241]. Similar results were shown for ASCs de-differentiated into the myCAF phenotype. These cells promoted TME fibrosis, desmoplasia and chemoresistance in a stroma-rich xenograft mouse model [245] and enhanced breast cancer cell invasion in vitro [94]. Given that obesity was reported to stimulate the de-differentiation of ASCs into an cancer-associated phenotype [126], this might explain that obesity fuels the malignant progression of breast cancer by reshaping its TME.

#### 5.2.3. De-Differentiation of MSCs/ASCs into iCAFs with Secretion of Soluble Factors and Exosomes

Another important CAF subpopulation is iCAFs with a low αSMA expression and high cytokine production as well as secretion [242], found in breast cancer and in pancreas ductal adenocarcinoma (PDAC) [240,253]. This extraordinary secretory activity is related to a high gene expression of important signaling regulators, such as *PDGFRα*, dermatopontin (*DPT*), C-type lectin domain family 3 member B (*CLEC3B*), collagen type XIV alpha 1 chain (*COL14A1*), lymphocyte antigen 6c1 (*Ly6c1*), hyaluronan synthase 1 (*HAS1*), *HAS2*, *IL6*, *IL8*, *IL11*, *CXCL1*, *CXCL2*, and *CCL2* [240,253]. The data from single-cell RNA sequencing of PDAC and bladder urothelial carcinoma tissue revealed that the cytokine-cytokine receptor interaction pathway was significantly enriched in iCAFs [243,255]. Increased gene levels of *VEGF*, *FGF*, *FGF7*, *IGF1,* and *IGF2*, known for their proliferation-promoting effects in endothelial, fibroblasts, and cancer cells, were also detected [243]. Indeed, the supernatant of iCAFs isolated from bladder urothelial carcinoma tissue promoted proliferation of tumor cells [243]. These cells were also capable of suppressing the immune response by interfering with the activity of CD8^+^ T-cells, CD4^+^ T-cells, Tregs, NK cells, mast cells, myeloid cells, and neutrophils through the secretion of various cytokines, such as CXCL1, CXCL12, CXCL16, IL6, IL8, IL11, IL33, LIF, PGE2, PVR cell adhesion molecule (PVR), podoplanin (PDPN), DPP4, PD1, PD2, and TGFβ [243,255]. This immunosuppressive function could be associated with the poor response to immunotherapy in fibrotic cancers with a high number of iCAFs [254]. Additionally, pharmacologic blockade or depletion of LIF, a key paracrine factor from iCAFs [250], was shown to reduce the progression of PDAC in a mouse model by modulating cancer cell differentiation and the EMT status [256]. Interestingly, the function of iCAFs was not limited to paracrine signaling. These cells expressed the genes *HAS1* and *HAS2* responsible for the synthesis of hyaluronan, which is a major component of the ECM [237], and its expression has been shown to correlate with low immune response and poor prognosis in multiple cancer entities, including breast cancer [257,258]. Furthermore, many factors released from iCAFs into the TME facilitate tumor growth, angiogenesis, and metastasis [259], though further in vivo and in vitro evidence for these functions is needed. Similar to myCAFs, the de-differentiation process of ASCs toward an iCAF phenotype could be recapitulated in the stroma-rich xenograft mouse model, which was found to be connected to increased tumor growth [245]. In accordance, a recent study showed that MSCs were able to de-differentiate into inflammatory cancer-associated cells by activating the IL1α/ETS-related transcription factor Elf-3 (Elf3)/yes1-associated transcriptional regulator (YAP) signaling axis [260]. These data strongly suggest that cancer-cell-educated ASCs/MSCs are capable of undergoing de-differentiation into iCAFs as well as myCAFs [245,260]; the latter may be driven by morbid obesity [126].

Collectively, these data highlight that breast cancer cells are able to de-differentiate ASCs/MSCs into different CAF subtypes (Figure 3). Multilateral communication between ASCs/MSCs, breast cancer cells, and the components of the TME promote breast cancer progression by activating various signaling pathways via paracrine signaling or direct cell–cell contact. 

## 6. Clinical Significance

ASCs/MSCs, in particular obese ASCs/MSCs, may contribute significantly to breast cancer development through several mechanisms, including remodeling the TME (“the soil for the seed”), promoting EMT, and inducing CSCs that cause clinical complications such as therapy resistance, cancer relapse, and metastasis. Moreover, mammary ASCs/MSCs may de-differentiate into CAFs, distribute at the interface between blood vessels and breast cancer cells, contribute to increased tumor interstitial fluid pressure, and represent a physical barrier to several drugs [251]. In fact, altered ECM induced by ASCs/MSCs/CAFs may induce tissue stiffness and increased tension, which have been associated with poor outcome in patients with many solid tumors [261]. Importantly, the immunosuppressive and poorly accessible TME drastically limits the potential of effective therapeutics. 

ASCs/MSCs/CAFs may also provide new therapeutic opportunities. Overcoming immunosuppression of the TME and suppressing the development of CSCs by targeting cancer-associated ASCs/MSCs/CAFs are of decisive importance for the effective treatment of breast cancer. Indeed, pre-clinical studies targeting ASCs by a killer peptide D-CAN showed promising results with a significantly reduced EMT and cancer progression in prostate cancer mouse models [210]. Moreover, a protein inducing apoptosis in CAFs and angiogenic endothelial cells by targeting a novel site of integrin αvβ3 displayed a strong reduction in intra-tumoral levels of EGF, IGF1, PDGF, collagen, and angiogenic vessels in an orthotopic xenograft model [262]. Consequently, malignant progression was highly decreased with reduced cancer cell proliferation, metastasis, tumor growth, and resistance to chemotherapy [262]. Other CAF targeted therapies include compounds against αFAP (myCAFs) [263], dasatinib/imatinib (PDGFR inhibitor, myCAFs) [264], galunisertib/vactosertib (TGFβRI inhibitor, myCAFs) [265], and ruxolitinib (JAK signaling inhibitor, iCAFs) [249]. Many of these inhibitors have been already analyzed in completed clinical trials [265]. In addition, ASCs/MSCs might also be an exceptional tool to deliver chemotherapeutics, demonstrated by a recent study where paclitaxel-loaded ASCs reduced breast tumor growth [266]. These interesting data highlight that targeting cancer-educated ASCs/MSCs/CAFs in the TME may pave a novel path to effectively combat malignancies, including breast cancer. 

## 7. Conclusions and Perspectives

Recent data clearly suggest that ASCs/MSCs, in particular obese ASCs/MSCs, play key roles in remodeling the TME and supporting breast cancer development. Much work remains. First, further investigations are required to unravel the complex crosstalk between ASCs/MSCs, breast cancer cells, and other stromal cells. In this context, breast cancer cells grown in 3D and co-cultured with ASCs/MSCs or other stromal cells will be useful to represent a more physiological morphology with prevalent cell junctions and polarity, and to resemble a more physiological phenotype in cell proliferation, gene expression, and differentiation [267]. Second, in-depth analyses are necessary to explore how obesity remodels the TME, affects the communication between breast cancer cells and ASCs/MSCs, and potentiates breast cancer cells to educate ASCs/MSCs into CAFs. Third, given the heterogeneity of ASCs/MSCs, additional work is required to identify and adequately classify various subpopulations that may have different functions in breast cancer progression, especially the subpopulation that is able to raise CSCs from breast cancer cells. This will be crucial to understand how ASCs/MSCs contribute to cancer development and may lead to the identification of new therapeutic targets or biomarkers as well as the use of ASCs/MSCs as therapeutic tools. In particular, the use of ASCs/MSCs as a. therapeutic tool could benefit from the in-depth characterization of different subtypes, as human umbilical-cord-derived MSCs have been discussed for their anti-tumorigenic effect [58,101]. Fourth, the future challenge is to elucidate the detailed molecular mechanisms in vivo by which obese ASCs/MSCs promote tumor growth, induce EMT, facilitate angiogenesis, raise CSCs, and fuel breast cancer metastasis. Finally, inhibition of the crosstalk between ASCs/MSCs and breast cancer cells could be an attractive strategy in cancer therapy. As ASCs/MSCs migrate toward cancer sites, it will also be interesting to develop ASCs/MSCs as a targeted anticancer therapy through genetic modification or engineering. 

## Figures and Tables

**Figure 1 cancers-14-03908-f001:**
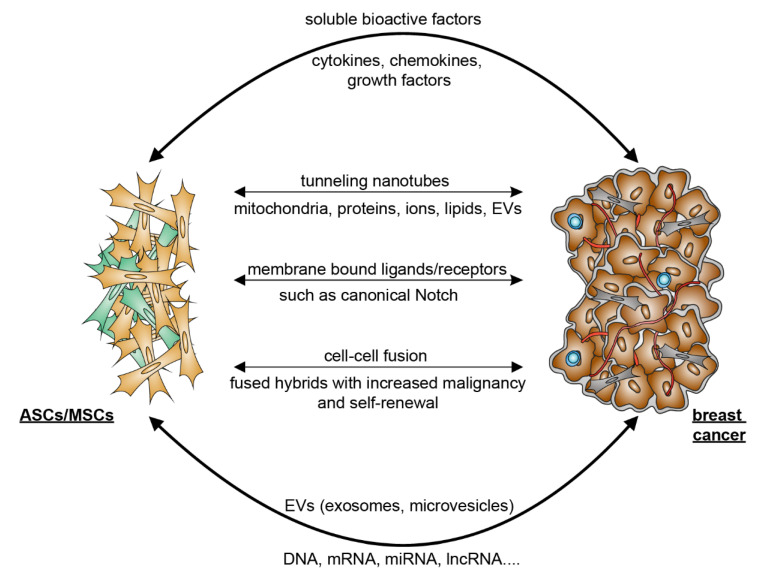
Simplified model representing the crosstalk between ASCs/MSCs and breast cancer cells. The communication between ASCs/MSCs and breast cancer cells may occur directly via cell–cell contact, namely TNTs, cell fusion, and the binding of membrane-bound ligands to receptors, or indirectly via released soluble bioactive factors such as cytokines, chemokines, and growth factors, and EVs including exosomes and microvesicles. EV, extracellular vesicles; ASCs, adipose tissue-derived mesenchymal stromal/stem cells; MSCs, mesenchymal stromal/stem cells; mRNA, messenger RNA; miRNA, microRNA; lncRNA, long non-coding RNA; TNTs, tunneling nanotubes.

**Figure 2 cancers-14-03908-f002:**
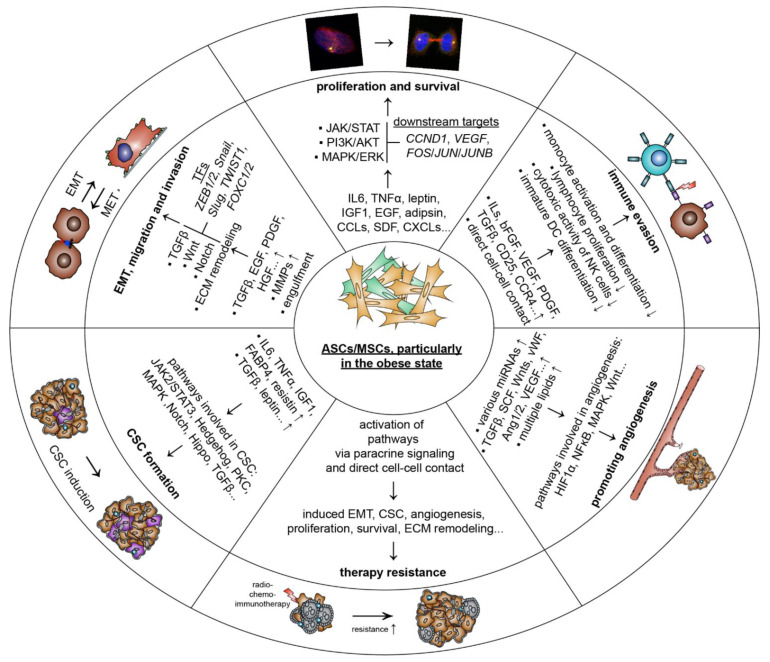
Schematic representation of potential effect of ASCs/MSCs on breast cancer cells and related molecular mechanisms. ASCs/MSCs may promote breast cancer cell proliferation and survival, EMT, migration, and invasion; CSC formation; angiogenesis; immune evasion; and therapy resistance. ASCs, adipose-tissue-derived mesenchymal stromal/stem cells; MSCs, mesenchymal stromal/stem cells; IL6, interleukin 6; EMT, epithelial-to-mesenchymal transition; STAT3, signal transducer and activator of transcription 3; ERK, extracellular-signal regulated kinase; IGF1, insulin-like growth factor 1; VEGF, vascular endothelial growth factor; MAPK, mitogen-activated protein kinase; AKT, protein kinase B; CCL2, monocyte chemotactic and activating factor; EGF, epithelial growth factor; PDGF-D, platelet-derived growth factor D; Wnt, wingless/integrated; TGFβ, transforming growth factor β; PI3K, phosphoinositide 3-kinase; CCR4, C-C motif chemokine receptor 4; TNFα, tumor necrosis factor α; CD25, cluster of differentiation 25; SNAI, snail family transcriptional repressor; ZEB1, zinc finger E-box binding homebox 1; CAF, cancer-associated fibroblast; CSC, cancer stem cell; JAK2, Janus kinase 2; FABP4, fatty acid binding protein 4; PKC, protein kinase C; HGF, hepatocyte growth factor; MMP, matrix metalloprotease; bFGF, basic fibroblast growth factor; vWF, von-Willebrand factor.

**Figure 3 cancers-14-03908-f003:**
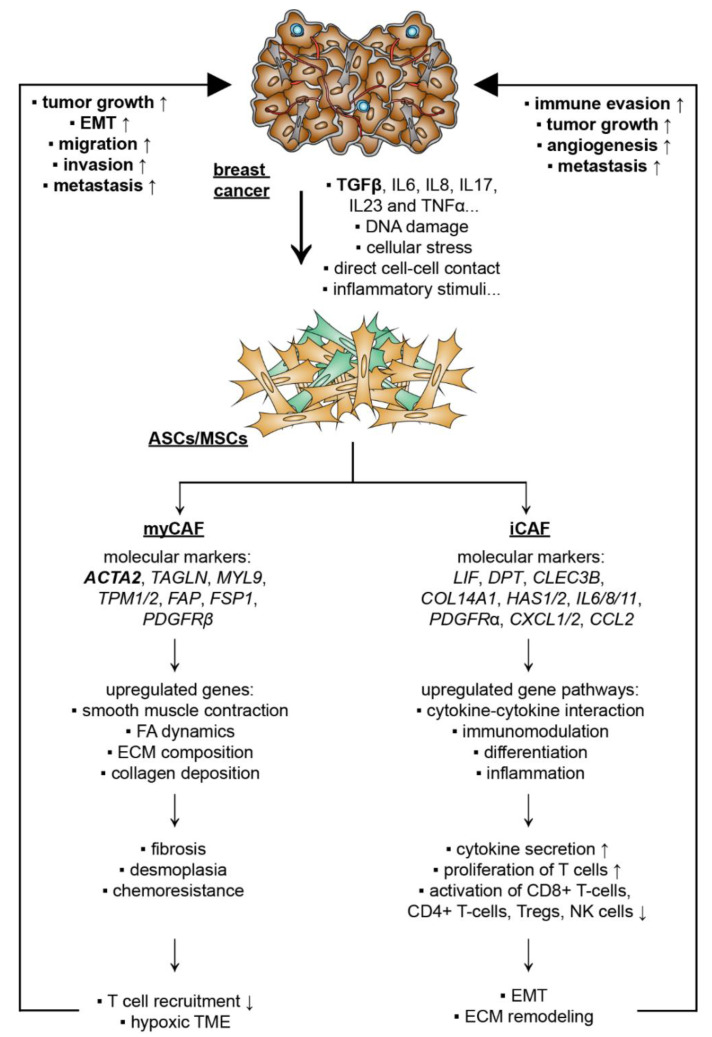
Simplified model showing that breast cancer cells induce de-differentiation of MSCs/ASCs into at least two distinct CAF subtypes. The de-differentiation process of MSCs/ASCs in the TME of breast cancer is triggered by multiple factors including cytokines TGFβ, IL6, IL8, IL17, IL23, and TNFα, DNA damage, cellular stress, direct cell–cell contact and inflammatory stimuli. This malignant transformation shifts ASCs/MSCs into several cancer-supportive populations including two typical phenotypes: myCAFs promoting tumor growth, EMT, migration, invasion, and metastasis, and iCAFs mediating immune evasion, tumor growth, angiogenesis, and metastasis. ASCs, adipose tissue-derived mesenchymal stromal/stem cells; MSCs, mesenchymal stromal/stem cells; TGFβ, transforming growth factor β; IL6, interleukin 6; EMT, epithelial-to-mesenchymal transition; TNFα, tumor necrosis factor α; ECM, extracellular matrix; TME, tumor microenvironment; CD8, cluster of differentiation 8; myCAF, myofibroblast cancer-associated fibroblast; iCAF, inflammatory cancer-associated fibroblast; ACTA2, smooth muscle actin, TAGLN, transgelin; MYL9, myosin light chain 9; TPM, tropomyosin; FAP, fibroblast activation protein; FSP1, fibroblast-specific protein-1; PDGFRβ, platelet-derived growth factor receptor beta; LIF, leukemia inhibitory factor; CXCL, chemokines C-X-C ligand; HAS, hyaluronan synthase; CCL, monocyte chemotactic and activating factor; COL14A1, collagen type XIV alpha 1 chain.

**Table 1 cancers-14-03908-t001:** Functional alterations of breast cancer cells induced by ASCs/MSCs.

ASC/MSC Source	Study Design	Functions and Molecular Mechanisms	Ref.
** Breast cancer promoting effects of ASCs/MSCs derived from human tissues **			
Human ASCs derived from visceral and subcutaneous adipose tissue	MCF7, MDA-MB-231 BC cell lines and MCF10A in vitro	Direct co-culture of ASCs promoted proliferation of BC cells with an upregulation of *AURKA*, *PLK1*, *BCL6*, IL6, and IL8, whereas indirect co-culture led to EMT of BC cells via STAT3 and ERK signaling.	[32]
Human ASCs obtained from ATCC	MCF7 and BT474 BC cell lines in vitro	Supernatant of ASCs increased BC cell proliferation and radiotherapy resistance by IGF1 secretion. BC cells overexpressed IGF1R upon radiotherapy.	[102]
Human BM-MSCs	MCF7, T47D, and SK-Br-3 BC cell lines in vitro	BM-MSC supernatant increased proliferation of BC cells independent of IL6 and VEGF, but both signaling proteins stimulated migration by the activation of MAPK, AKT, and p38 MAPK.	[103]
Human BC-derived MSCs	MCF7 BC cell line in vitro	Mammary MSCs increased proliferation and cisplatin resistance of MCF7 cells by triggering the IL6/STAT3 pathway.	[104]
Human MSCs from primary BC tissue	Co-transplantation BC xenograft mouse model of MCF7 and MSCs in vivo and in vitro	Mammary MSCs promoted BC proliferation and mammosphere formation via EGF/EGFR/AKT signaling.	[105]
Human ASCs from adipose tissues	MCF-7, BT-474, T-47D, and 4T1 BC cell lines in vitro and in vivo	PDGF-D secreted by ASCs stimulated tumor growth in vivo, mammosphere formation in vitro, and EMT in BC cells.	[106]
Human MSCs from supraclavicular lymph node (LN-MSCs) and liver (Lv-MSCs)	MDA-MB-231, –436, –468, MCF7 BC cell lines, and MCF10A cells in vitro and in vivo	The engulfment of MSCs by BC cells increased the gene expression of *WNT5A*, *MSR1*, *ELMO1*, *IL1RL2*, *ZPLD1*, and *SIRPB1.* This further increased BC cell migration, invasion, and mammosphere formation in vitro and the tumor metastasis in vivo.	[107]
Human ASCs from facial or abdominal liposuction	MCF7 BC cell line in vitro	ASCs co-cultured with MCF7 stimulated EMT in BC cells. The data also suggest that EMT was induced by the cross-interactions with the TGFβ/Smad and PI3K/AKT pathways.	[108]
Human ASCs isolated from SAT via bariatric surgery, and mammary ASCs from subcutaneous breast preadipocytes	MCF7 and SUM149 BC cell lines in vitro, and orthotopic grafting of 4T1 cells into the mammary fat pad in vivo	Both ASCs subtypes suppressed the cytotoxicity of cisplatin and paclitaxel. Depletion of ASCs by D-CAN, a proapoptotic peptide targeting specific ASCs, reduced spontaneous BC lung metastases in a mouse allograft model and a BC xenograft model, when combined with cisplatin treatment.	[109]
Human ASCs isolated from breast adipose tissues of breast cancer patients and normal individuals underwent cosmetic mammoplasty surgery	Breast tissue and BC tissue samples in vitro	ASCs isolated from breast cancer patients displayed elevated levels of IL10 and TGFβ1, and the supernatant stimulated the expression of IL4, TGFβ1, IL10, CCR4, and CD25 in PBLs.	[110]
Human ASCs isolated from breast tumor (T-MSC) and normal breast adipose tissue (N-MSC)	Breast tissue and BC tissue samples in vitro, PBLs in vitro	The TME altered the secretome of T-MSCs with increased secretion of TGFβ, PGE2, IDO, VEGF, and lowered secretion of MMP2/9 compared to N-MSCs. T-MSCs also stimulated the proliferation of PBLs.	[111]
Human ASCs isolated from normal breast adipose tissue (nASCs) or that of a woman with breast cancer (cASCs)	Breast tissue and BC tissue samples in vitro, B cells and Tregs in vitro	nASCs reduced proliferation of B cells in direct co-culture, and the TNFα^+^/IL10^+^ B cells ratio decreased in all co-cultures with ASCs, to a barely significantly higher extent in cASCs. nASCs shifted the cytokine profile of B cells toward an anti-inflammatory profile.	[112]
Human ASCs isolated from the breast adipose tissue of reduction mammoplasty patients with different BMI	MCF7 and SUM159PT BC cell lines and HMEC breast cell line in vitro	Supernatant of all analyzed ASCs stimulated proliferation, migration, and invasion of breast cancer cells and increased the number of lipid droplets in their cytoplasm. This was mechanistically associated with the upregulated expression of the fatty acid receptor CD36, presenting the capacity of ASCs to induce metabolic reprogramming via CD36-mediated fatty acid uptake.	[113]
Human primary subcutaneous pre-adipocytes (pre-hASCs, Lonza)	MCF7, T47D, ZR-75-1, SK-BR-3 BC cell lines and murine 3T3-L1 pre-adipocytes in vitro	Conditioned medium of ASCs stimulated proliferation and migration of MCF7, T47D, SK-BR-3, and ZR-75-1 cells. Additionally, supernatant of ASCs upregulated the expression of S100A7 and its knockdown abrogated the tumorigenic effect of ASCs on the tested breast cancer cells.	[114]
** Breast cancer promoting effects of ASCs/MSCs derived from murine tissue **			
Murine MSCs derived from spontaneous lymphomas, mouse bone marrow, and mouse ears	Syngeneic tumor transplantation mouse model in vivo	TNFα dependent monocyte/macrophage recruitment led to increased tumor volume upon co-injection with MSCs, associated with CCR2 dependent immunosuppression of neutrophils, monocytes, and macrophages.	[115]
Murine BM-MSCs and MSCs isolated from murine lung cancers	4T1 BC mouse model in vivo	BM-MSCs and MSCs from lung cancers were able to recruit CXCR2^+^ neutrophils into the tumor by TNFα via activation of CXCL1, CXCL2, and CXCL5 and promoted tumor metastasis.	[116]
Murine BM-MSCs	Murine mammary cancer cell lines PyMT-Luc, 17LC3-Luc and LLC in vitro	Secretion of CXCL5 by BM-MSCs increased, but without significance, while proliferation of murine BC cell lines was unchanged, whereas CXCL1 and CXCL5 promoted BC cell migration.	[117]
Murine and human BM-MSCs	4T1 BC mouse model in vivo and in vitro	Both types of BM-MSCs stimulated 4T1 BC cell proliferation in vivo and in vitro upon direct cell–cell contact. BM-MSCs also promoted vessel formation of HUVECs in vitro and in vivo in DU145 tumors via TGFβ, VEGF, and IL6 release.	[118]
Murine ASCs isolated from abdominal cavity	4T1 BC mouse cell line in vitro and CT26 murine colon cancer cell line in vitro	Co-culture of ASCs induced stemcellrelated genes in cancer cells such as *SOX2*, *NANOG*, *ALDH1,* and *ABCG2*. ASCs accelerated tumor growth. Secretion of IL6 regulated stemcellrelated genes and activated JAK2/STAT3 in murine cancer cells.	[119]
** Breast cancer promoting effects of obese ASCs/MSCs **			
Human ASCs isolated from breast cancer tissue of lean and obese patients	Human BC patient-derived xenograft cells in vivo	Adipsin secreted by obese ASCs stimulated factor B and C3a, which induced BC proliferation and expression of CSC genes *CD44*, *CXCR4*, *SNAI2*, *SNAI1*, *ZEB1,* and *BMI1.*	[120]
Human lean and obese ASCs isolated from abdominal lipo- aspirates of subcutaneous adipose tissue	MCF7, ZR75, or T47D BC cell lines in vitro and MCF7 xenograft mouse model in vivo	Leptin secreted from obese ASCs enhanced BC proliferation and increased the expression of EMT and metastasis-related genes such as *Serpine1*, *MMP2,* and *IL6*.	[121]
Human lean (ln) and obese (ob) ASCs from abdominal lipo- aspirates of subcutaneous adipose tissue	MCF7 and MDA-MB-231 BC cell lines in vitro	Increased proliferation of BC cells by leptin expression via estrogen stimulation and increased protein levels of CDKN2A, GSTP1, PGR, and ESR1 in BC cells co-cultured with ob-ASCs.	[122]
Human and murine ASCs isolated from lean and obese individuals	Tumor and stromal cell transplantation in a mammary mouse xenograft model in vivo and MCF7 BC cell line in vitro	Obese ASCs secreted higher levels of IGF1, promoting tumor growth and metastasis, which could be partially ameliorated by weight loss.	[123]
Human lean and obese ASCs from abdominal lipoaspirates of subcutaneous adipose tissue	BT20, MDA-MB-231, MDA-MB-468, MCF7, and HCC1806 BC cell lines in vitro and patient-derived xenograft mouse model	Obesity increased the tumorigenic capacity of ASCs indicated by increased EMT genes *Serpine1*, *SNAI2,* and *TWIST1*. This effect was likely mediated via leptin, since its knockdown led to reduced pro-metastatic effects of obese ASCs.	[124]
Human ASCs isolated from lipoaspirate of subcutaneous adipose tissue from lean and obese patients.	MCF7, T47D, and ZR-75 BC cell lines in vitro	Obese ASCs induced a cancer-stem-like phenotype in BC cells with elevated gene expression of *Notch1*, *Notch3*, *DLL1,* and *JAG2*. This led to radioresistance and reduced oxidative stress after radiation in co-cultured BC cells mediated by leptin.	[83]
Human lean and obese ASCs derived from mammary adipose tissue	MDA-MB231 BC cell line and MCF10AT1 in vitro	Obese ASCs activated BC cell migration more effectively compared to lean ASCs by direct co-culture. Obese ASCs had an increased potential for ECM remodeling.	[125]
Human lean and obese ASCs from abdominal lipoaspirates of subcutaneous adipose tissue	MCF7 BC cell line in vitro	The known CAF markers *NG2, ACTA2, VEGF, FAP,* and *FSP* were elevated in obese ASCs. Obese ASCs were more potent in inducing the gene expression of pro-tumorigenic factors in BC cells including *Serpin1, CCL5, TARC (CCL17*)*, IL24, IL6, IGFBP3, adiponectin,* and *leptin*.	[126]
Human lean and obese ASCs isolated from elective liposuction	MCF7 BC cell line in vitro and patient-derived mammary xenograft (PDX) mouse model in vivo	The increased tumor growth rate observed in obese-ASCs-enriched PDX tumors was leptin dependent. The increased metastatic capacity was leptin independent and was associated with increased gene expression of *Serpine1* and *ABCB1* in tumor cells.	[127]

Abbreviations: ASCs, adiposetissue-derived mesenchymal stromal/stem cells; MSCs, mesenchymal stromal/stem cells; BM-MSCs, bonemarrow-derived mesenchymal stromal/stem cells; IL6, interleukin 6; EMT, epithelial-to-mesenchymal transition; BC, breast cancer; STAT3, signal transducer and activator of transcription 3; ERK, extracellular-signal regulated kinase; IGF1, insulin-like growth factor 1; IGF1R, insulin-like growth factor 1 receptor; VEGF, vascular endothelial growth factor; MAPK, mitogen-activated protein kinase; AKT, protein kinase B; EGF, epithelial growth factor; EGFR, epithelial growth factor receptor; PDGF-D, platelet-derived growth factor D; WNT5A, wingless/integrated 5a; MSR1, macrophage scavenger receptor types I; ELMO1, engulfment and cell motility protein 1; IL1RL2, interleukin 1 receptor like 2; AURKA, Aurora kinase A; PLK1, Polo-like kinase 1; BCL6, B-cell lymphoma 6; SAT, subcutaneous adipose tissue; CDKN2A, cyclin-dependent kinase inhibitor 2A; GSTP1, glutathione S-transferase P; ABCB1, ATP-binding cassette subfamily B member 1; ZPLD1, zona pellucida-like domain-containing 1; SIRPB1, signal-regulatory protein beta-1; TGFβ, transforming growth factor β; Smad, suppressor of mothers against decapentaplegic family member; PI3K, phosphoinositide 3-kinase; CCR4, C-C motif chemokine receptor 4; PBL, peripheral blood lymphocytes; MMP, matrix metalloprotease; PGE2, prostaglandin E2; IDO, indoleamine 2,3-dioxygenase; TNFα, tumor necrosis factor α; CXCL1, C-X-C motif chemokine ligand 1; HUVEC, human umbilical vein endothelial cell; CD44, cluster of differentiation 44; SNAI, snail family transcriptional repressor; ZEB1, zinc finger E-box binding homebox 1; BMI, body mass index; PGR, progesterone receptor; ESR1, estrogen receptor 1; ob, obese; ln, lean; DLL1, delta-like canonical Botch ligand 1; JAG2, jagged canonical Notch ligand 2; IGFBP3, insulin-like growth factor binding protein 3; JAK2, Janus kinase.

**Table 3 cancers-14-03908-t003:** Subtypes of cancer associated ASCs/CAFs in breast cancer.

Fibroblast/CAF Source	Study Design	Functions and Molecular Mechanisms	Ref.
Human adipose progenitors (APs) isolated from adipose tissue and breast-APs (B-APs) isolated from breast adipose tissue	MCF-7 and T47D cell lines in vitro	Primary cilia of APs were required for de-differentiation of APs into CAFs stimulated by breast cancer cells. Inhibition of cilia stopped the malignant transition of APs. Primary cilia mediated TGFβ1 signaling to APs.	[234]
Human lean and obese ASCs from abdominal lipoaspirates of subcutaneous adipose tissue	MCF7 cell line in vitro	Co-culture of breast cancer cells with lean and obese ASCs induced a CAF-like phenotype with elevated gene expression of *NG2, ACTA2, VEGF, FAP,* and *FSP.* This cancer-educated phenotype was enhanced in obese ASCs compared to lean counterparts. Obese ASCs were more potent in inducing the expression of pro-tumorgenic factors in breast cancer cells including *Serpin1, CCL5, TARC, IL24, IL6, IGFBP3, adiponectin,* and *leptin*.	[126]
Human adipocytes/pre-adipocytes isolated from breast cancer tissue or reduction mammoplasty	Co-culture with murine 3T3-F442A pre-adipocytes cell line, murine 4T1 breast cancer cell line, human breast cancer cell line SUM159PT in vitro	Co-culture of breast cancer cells with mature adipocytes or pre-adipocytes led to enhanced secretion of fibronectin and collagen I. This was associated with enhanced migration/invasion and the expression of known CAF marker FSP1. The de-differentiation process was triggered by the reactivation of the Wnt/β-catenin pathway in response to Wnt3a.	[251]
Human ASCs isolated from unprocessed subcutaneous adipose tissue	MDA-MB-231 and MCF7 cell lines and supernatant, in vitro	ASCs were de-differentiated in response to supernatant of breast cancer cells, shown by the expression of *ACTA2, SDF1, CCL5,* and *tenascin-C,* mediated by TGFβ1/Smad3.	[94]
Immortalized human AD-MSC cell line ASC52telo (ATCC)	Capan-1 and MIAPaCa-2 human PDAC cell lines and stroma-rich cell-derived xenograft (Sr-CDX) mouse model in vitro/in vivo	The SR-CDX model resembled the PDAC phenotype induced by CAFs with accelerated tumor growth, stromal cell proliferation, chemoresistance, and dense stroma. Single-cell RNA sequencing revealed that the CAFs in the TME were derived from the transplanted AD-MSCs, which de-differentiated into known and unknown CAF subtypes.	[245]
Data sets from multiple pan-cancer biopsy tissues	Single-cell RNA sequencing data sets from multiple cancer biopsies to recapitulate ASC de-differentiation process in vitro	This analysis revealed that CAFs originated from a particular subset of ASCs present in the stroma vascular fraction of normal adipose tissue. The transition stages of ASCs were recapitulated toward a cance-associated phenotype by using a rich pancreatic cancer dataset. At the endpoint of this transition process, the cells presented the following upregulated genes: *MMP11, COL11A1, C1QTNF3, CTHRC1, COL12A1, COL10A1, COL5A2, THBS2, AEBP1, LRRC15,* and *ITGA11.*	[246]
Immortalized human AD-MSC cell line ASC52telo (ATCC)	Capan-1, SUIT-2, and MIAPaCa-2 human PDAC cell lines and stroma-rich cell-derived xenograft (Sr-CDX) mouse model in vitro	AD-MSCs acted as precursors for CAFs in vitro. AD-MSCs could be induced into myCAFS and iCAFs upon co-culture with PDAC cells. Direct co-culture led to a myCAF phenotype, whereas indirect co-culture induced an iCAF gene expression pattern.	[247]
Human ASCs (ADSC-GM) from Lonza	MDA-MB-231 breast cancer cell line and HUVECs in vitro	EVs from MDA-MB-231 converted ASCs into a myCAF-like phenotype, with increased VEGF and ECM remodeling, and partly driven by MAPK signaling.	[252]

Abbreviations: AP, adipose progenitors; ASCs, adipose-tissue-derived mesenchymal stromal/stem cells; MSCs, mesenchymal stromal/stem cells; BM-MSCs, bone-marrow-derived mesenchymal stromal/stem cells; IL6, interleukin 6; CAF, cancer-associated fibroblast; myCAF, myofibroblast cancer-associated fibroblast; EMT, epithelial-to-mesenchymal transition; VEGF, vascular endothelial growth factor; MAPK, mitogen-activated protein kinase; AKT, protein kinase B; EGF, epithelial growth factor; BC, breast cancer; C1QTNF3, complement C1q tumor necrosis-factor-related protein 3; CTHRC1, collagen triple helix repeat-containing protein 1; THBS2, thrombospondin-2; AEBP1, adipocyte enhancer-binding protein 1; LRRC15, leucine-rich repeat-containing protein 15; ACTA2, actin alpha 2; MMP, matrix metallopeptidase.

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
