# Peer review of "Adipose Tissue-Derived Mesenchymal Stromal/Stem Cells, Obesity and the Tumor Microenvironment of Breast Cancer"

_cancers, 2022, doi:10.3390/cancers14163908_

Round 1
Reviewer 1 Report
The review “Adipose tissue-derived mesenchymal stromal/stem cells, obesity and the microenvironment of breast cancer” by Ritter et al. provides a comprehensive and detailed description of the characteristics and role of mesenchymal/stromal cells in breast cancer. This review is well written presenting clear tables and figures.
Minor mistakes
I believe that a typo error occurs in the breast cancer gene 1 abbreviation.
Similarly Serpine1.
Pag 14, line 382: delete “were”
Paragraphs 4.1.3 and 4.1.4 have the same title “Escape of immune response”.
Figure 2: rotate immune evasion
Author Response
We thank the reviewers and editors for their comments and suggestions, which have helped us to improve our manuscript. The following is our response to the comments, point-by-point (A for answer, in blue and italic).
Reviewer #1:
The review “Adipose tissue-derived mesenchymal stromal/stem cells, obesity and the microenvironment of breast cancer” by Ritter et al. provides a comprehensive and detailed description of the characteristics and role of mesenchymal/stromal cells in breast cancer. This review is well written presenting clear tables and figures.
Minor mistakes
I believe that a typo error occurs in the breast cancer gene 1 abbreviation.
A: Thanks for the comment, it is changed to BRCA1.
Similarly Serpine1.
A: It is changed from serpin1 to serpine1.
Pag 14, line 382: delete “were”
A: The additional “were” is deleted.
Paragraphs 4.1.3 and 4.1.4 have the same title “Escape of immune response”.
A: The title of chapter 4.1.4 is changed to “Inducing EMT, migration and invasion”.
Figure 2: rotate immune evasion
A:. The immue evasion in Figure 2 is now rotated.
Reviewer 2 Report
In this review article, Ritter et al. summarized the correlation of adipose tissue-derived mesenchymal stromal/stem cell to breast cancer, and in particular how obesity impacts the microenvironment of cancer cell and ASCs/MSCs crosstalk. The manuscript listed the different means of ASCs/MSCs and cancer cell interaction, the different influence of ASCs/MCSs to cancer-associated molecular mechanisms: promoting proliferation, angiogenesis, EMT process, cancer stem cell, as well as escaping immune response, and facilitating therapy resistance. The authors further discussed how cancer cells influence ASCs/MSCs in de-differentiation into fibroblasts. The discussion provides a broad understanding of the topic, and is balanced in its contents. The studies cited are timely, and suggests directions for future research. One question here regards the comment in line 292: what are some examples of “anti-tumorigenic” effects from recent studies, as this could be interesting to be included in the discussion. Also, I believe the sections are numbered incorrectly since section 4, and the current 4.1.4 has an incorrect heading.
Author Response
Reviewer #2:
In this review article, Ritter et al. summarized the correlation of adipose tissue-derived mesenchymal stromal/stem cell to breast cancer, and in particular how obesity impacts the microenvironment of cancer cell and ASCs/MSCs crosstalk. The manuscript listed the different means of ASCs/MSCs and cancer cell interaction, the different influence of ASCs/MCSs to cancer-associated molecular mechanisms: promoting proliferation, angiogenesis, EMT process, cancer stem cell, as well as escaping immune response, and facilitating therapy resistance. The authors further discussed how cancer cells influence ASCs/MSCs in de-differentiation into fibroblasts. The discussion provides a broad understanding of the topic, and is balanced in its contents. The studies cited are timely, and suggests directions for future research. One question here regards the comment in line 292: what are some examples of “anti-tumorigenic” effects from recent studies, as this could be interesting to be included in the discussion.
A: Thanks for the suggestion. We have added the anti-tumorigenic effects (reduced angiogenesis, tumor growth and migration/invasion with additional literature in the main text (page 8, lines 294-300) and discussion (page 32, lines 845-848).
Also, I believe the sections are numbered incorrectly since section 4, and the current 4.1.4 has an incorrect heading. ? ? ?
A: We are sorry. The title of chapter 4.1.4 is changed to “Inducing EMT, migration and invasion”. The title of section 4 “4. Mutual interaction between ASCs/MSCs and breast cancer cells” is correct.
Reviewer 3 Report
The manuscript by Ritter is a review on the effect of obesity on the development of breast cancer. Taking the adipose tissue as the central point of the microenvironment of breast cancer, they explain in a brief introduction the prevalence of obesity as well as the diseases associated to it. In addition, they associate inflammation and adipose-derived mesenchymal stromal/stem cells (ASCs/MSCs) as crucial contributors to the progression of breast cancer.
The manuscript is well organized and easily readily. Ritter et al., make a wide revision (22 year and 265 references) on the relation of obesity and breast cancer deepening on the signaling pathways involve on cell proliferation and survival, angiogenesis, migration and invasion, the epithelial-mesenchymal transition, cancer stem cell development, immune evasion, therapy resistance, and the potential impact of breast cancer cells on ASCS/MSCs.
They include a section of clinical significance and other one of conclusions and perspectives.
Author Response
A: Thanks for the kind comments.